# Enhancement Effects and Mechanism Studies of Two Bismuth-Based Materials Assisted by DMSO and Glycerol in GC-Rich PCR

**DOI:** 10.3390/molecules28114515

**Published:** 2023-06-02

**Authors:** Zhu Yang, Junlei Yang, Lihuan Yue, Bei Shen, Jing Wang, Yuqing Miao, Ruizhuo Ouyang, Yihong Hu

**Affiliations:** 1Institute of Bismuth and Rhenium Science, School Materials and Chemistry, University of Shanghai for Science and Technology, Shanghai 200093, China; 203612304@st.usst.edu.cn (Z.Y.);; 2Institut Pasteur of Shanghai, Chinese Academy of Sciences, University of Chinese Academy of Sciences, Shanghai 200031, China; jwang1@ips.ac.cn; 3CAS Key Laboratory of Molecular Virology & Immunology, Institutional Center for Shared Technologies and Facilities, Pathogen Discovery and Big Data Platform, Institut Pasteur of Shanghai, Chinese Academy of Sciences, Shanghai 200031, China; lhyue@ips.ac.cn (L.Y.); bshen@ips.ac.cn (B.S.); 4School of Life Sciences and Biotechnology, Shanghai Jiao Tong University, Shanghai 200240, China

**Keywords:** PCR, bismuth-based materials, GC rich, enhance, mechanism

## Abstract

Polymerase chain reaction (PCR) has extensive bioanalytical applications in molecular diagnostics and genomic research studies for rapid detection and precise genomic amplification. Routine integrations for analytical workflow indicate certain limitations, including low specificity, efficiency, and sensitivity in conventional PCR, particularly towards amplifying high guanine–cytosine (GC) content. Further, there are many ways to enhance the reaction, for example, using different PCR strategies such as hot-start/touchdown PCR or adding some special modifications or additives such as organic solvents or compatible solutes, which can improve PCR yield. Due to the widespread use of bismuth-based materials in biomedicine, which have not yet been used for PCR optimization, this attracts our attention. In this study, two bismuth-based materials that are inexpensive and readily available were used to optimize GC-rich PCR. The results demonstrated that ammonium bismuth citrate and bismuth subcarbonate effectively enhanced PCR amplification of the GNAS1 promoter region (∼84% GC) and APOE (75.5% GC) gene of Homo sapiens mediated by Ex Taq DNA polymerase within the appropriate concentration range. Combining DMSO and glycerol additives was critical in obtaining the target amplicons. Thus, the solvents mixed with 3% DMSO and 5% glycerol were used in bismuth-based materials. That allowed for better dispersion of bismuth subcarbonate. As for the enhanced mechanisms, the surface interaction of PCR components, including Taq polymerase, primer, and products with bismuth-based materials, was maybe the main reason. The addition of materials can reduce the melting temperature (T_m_), adsorb polymerase and modulate the amount of active polymerase in PCR, facilize the dissociation of DNA products, and enhance the specificity and efficiency of PCR. This work provided a class of candidate enhancers for PCR, deepened our understanding of the enhancement mechanisms of PCR, and also explored a new application field for bismuth-based materials.

## 1. Introduction

PCR is widely used in DNA and RNA molecular analysis and diagnosis. However, amplifying the high GC content of DNA fragments is more difficult than for non-high GC content target fragments [1]. GC-rich sequences in the human genome are common and include important regulatory domains such as promoters, boosters, and control elements [2,3]. PCR amplification of GC-rich DNA is often problematic because the secondary structure of DNA is stable and not easy to melt. These secondary structures cause DNA polymerases to stop, leading to incomplete and non-specific amplification [4]. Several methods have been developed to amplify these problematic GC-rich DNA fragments. The reaction mixture, which includes organic molecules such as dimethyl methylene (DMSO), glycerol, polyethylene glycol, methylene-amide, beetroot, 7-deaza GTP, and dUTP, has been shown to improve the amplification of GC-rich DNA sequences [5,6,7,8,9]. In addition, different PCR strategies have been improved, such as thermal start PCR by gradually decreasing PCR and thermal starting methods combined with touch to increase PCR production [10]. However, these optimization effects are confined, limiting their wide application. In the past two decades, with the wide application of nanomaterials, many nanomaterials such as gold nanoparticles (Au NPs), graphene, carbon nanotubes, titanium dioxide and quantum dots have been used for PCR amplification [11]. They can not only effectively enhance amplification specificity, but also increase the yield. However, to successfully synthesize nanomaterials with excellent performance, it is often extremely complex, requiring multiple surface modifications and adjustment of synthesis parameters. Therefore, finding new materials with comprehensive optimization effects and simple preparation methods at low prices would be highly desirable.

Among the existing materials, bismuth-based materials have broad prospects for biomedical applications [12,13,14]. First of all, the storage of bismuth in China accounts for its first place in the world, which is a cheap metal; secondly, the bismuth complex shows good photoelectric signal conversion performance and stability; in addition, the good photothermal conversion efficiency of bismuth-based materials makes it suitable for tumor treatment [15,16]; finally, bismuth, which has a good antibacterial effect, has been widely used in the treatment of Helicobacter pylori and has also been used to treat gastric ulcers and to prevent and treat of diarrhea [17,18]. Bismuth’s non-toxic properties and good biocompatibility allow its application in molecular biology. Even though the applications of bismuth-based materials have received widespread attention, it has yet to be studied in terms of enhancing the PCR effect and its mechanism.

The mechanism of nano-PCR is still unclear because of the PCR reaction system’s complexity and materials’ characteristics. The possible mechanisms are as follows: (1) surface interactions between materials and PCR components [19,20,21,22,23]; (2) thermal conversion of materials rate [24,25]; (3) electrostatic interaction [26,27,28]; (4) analogous to ssDNA binding protein (SSB) [22]; (5) catalytic activity [29]. There is no doubt that these mechanisms cannot explain the impact on all the materials in PCR, and more undiscovered mechanisms need to be explored.

We used ammonium bismuth citrate and bismuth subcarbonate to optimize high GC-rich PCR. The bismuth-based compounds are cheap, readily available, and non-toxic, and bismuth subcarbonate has good medical value. They were all dissolved in a mixture of DMSO and glycerol in a certain ratio, which solves the problem that bismuth subcarbonate is insoluble in water. We found that a weak target band appeared after the mixed solvent was added alone, and the specific band within the proper concentration range was significantly enhanced after adding bismuth-based materials on the former basis. Based on the physicochemical properties of the two bismuth-based materials, we suspect their enhanced effect on PCR may be due to the first point summarized in the above mechanisms, which is surface interactions between materials and PCR components. Typical components in PCR include polymerases, primers, templates, and products. Therefore, we carried out a series of mechanism research experiments such as conventional PCR, gel electrophoresis, real-time PCR, and melting curve, and the hypothesis was confirmed. This simple, efficient, low-cost approach should find broad applications in molecular genomics.

## 2. Results and Discussion

### 2.1. Optimization of Conditions for PCR Systems

(1)Optimization for amplifying GNAS1 promoters

The GC content in the GNAS1 gene is as high as about 84%, which affects the optimal annealing temperature and primer specificity, making PCR amplification more difficult [30]. Although 7-desnozo-2′ deoxyguanosine [31] is effective for amplification of GNAS1 gene promoters, a rather time-consuming "deceleration" PCR procedure based on "touchdown" PCR is still critical [32,33]. Here, we optimized the system with different proportions of the enzyme, DMSO and glycerol, along with eight groups of experimental conditions optimization (see Table 1 for details). At the same time, the concentration of Mg^2+^ affects the activity of polymerase in the PCR system, as well as the melting temperature of the template and product, greatly influencing the reaction efficiency. So, we carried out the reaction with the concentration of Mg^2+^ at 1.5 mM and 2.0 mM. As shown in Figure 1, when treating 1.5 mM Mg^2+^, the amplification band (test 6) contained 2.50U enzyme, 3% DMSO, and 10% glycerol, which was the brightest, indicating that the target products were the highest. The target gene was amplified in all groups except for test 5, and test 1 had the least target product with an intensity of 33 (Figure 1b). When treating 2 mM Mg^2+^, weak target bands appeared in both test 5 and test 7 and non-specific bands also existed. So, none of the eight groups of experiments amplified the target bands with good specificity. This may be due to the high concentration of Mg^2+^, which reduces the specificity of the reaction (Appendix A).

DMSO is mainly used for PCR amplification in a GC-rich system. The possible mechanism is to improve the deformation of DNA with high GC content and reduce its secondary structure, so that polymerase extends the secondary structure to improve the specificity of PCR. Conventional PCR and RT-qPCR enzyme formulations extensively use glycerol (up to 50%) to stabilize the enzyme. As a cryoprotectant, it can prolong the stability time of the reagent. In order to make the control group with only DMSO and glycerol added to the experimental group more obviously contrasted with the experimental group with the addition of materials and to maximize the enhancement of the reaction amplification by materials, we preferred to choose an experimental group that could efficiently amplify genes with high GC content while saving reagents. Thus, considering the cost of the enzyme, the practicality of the enhancer combination, and the amplification effect of genes with high GC content, we chose 1.25U of enzyme amount, 3% DMSO, and 5% glycerol (test 1) at 1.5 mM Mg^2+^ concentration for subsequent amplification experiments.

(2)Optimization of the System for Amplifying the APOE Gene

PCR amplification of the APOE gene mediated by Ex Taq DNA polymerase was performed in two systems containing 0.4 μM primers, 2.0U polymerase (Appendix A) and 0.2 μM primers, 1.25U polymerase (Appendix A), while three different sources of g-DNA from RD cells, U87-MG cells, and cwbiotech Co., Ltd were studied in each system. As shown in Appendix A, both systems with different concentrations with three different sources of g-DNA amplified APOE genes have significant target bands, indicating that APOE genes are expressed in various cells. Follow-up experiments were performed with as few enzymes and primers as possible, selecting 0.2 μM primers, 1.25U polymerase, and g-DNA extracted from U87-MG cells for the reaction.

In addition, for the procedure of amplifying the APOE gene, the effect of shortening the annealing time and reducing the number of cycles on amplification was explored to maximize efficiency. The optimal annealing temperature was set to 60 °C according to the T_m_ value of the primers, and PCR procedures for amplifying GNAS1 and APOE gene were performed with different annealing times and cycle numbers under the additives (3% DMSO and 5% glycerol). It can be seen from Appendix A that when the annealing time was 45 s, more target products were amplified by 35 cycles, but there were some non-specific bands. When the annealing time was shortened to 30 s, the number of cycles was reduced to 30, and when the extension time was shortened to 20 s, the PCR time was shortened by 24 min. The amplification product was slightly reduced. Moreover, the non-specific products were removed. Here, we shortened the amplification reaction time, which had a certain impact on the yield of products but did not affect the gene we detected. We could still accurately determine the target gene and its size and eliminate the influence of non-specific products. So, the subsequent amplification of APOE gene was carried out with a program of 30 s annealing and 20 s extension for 30 cycles.

### 2.2. Validation of Amplification Products

The successful amplification of the GNAS1 promoter and APOE gene was verified by Sanger sequencing. Briefly, the loading solution of test 1 in Figure 1 and lane 1 in Appendix A was sent to the company (Biosune Biotechnology Co., Ltd., Shanghai, China) for sequencing, and the results of the DNA sequences are shown in Appendix A. The BLAST comparison of sample sequencing results is summarized in Appendix A. The amplified GNAS1 and APOE genes are the target products and can be studied for subsequent experiments.

### 2.3. Bismuth-Based Materials Enhanced PCR Amplification 

#### 2.3.1. Amplification of the GNAS1 Gene

To test the ability of bismuth-based compounds to enhance GC-rich PCR amplification, we first selected intact GNAS1 promoters in the human genome as templates [34]. The GNAS1 promoter contains a very GC-rich region with a regional GC content of up to 86%, and failed attempts to amplify this region have been reported [32]. We first investigated the effects of ammonium bismuth citrate and bismuth subcarbonate on the amplification of the GNAS1 promoter mediated by Ex Taq DNA polymerase.

The experiment of amplifying the GNAS1 promoter found that by adding 0.022 nM~22 nM (lanes 6–9) ammonium bismuth citrate, the reaction can effectively be enhanced (Figure 2a). The net optical density (NOD) in Figure 2b indicates the products in the band of the gel (minus the interference of the background in the gel image for the analysis). While the concentrations of ammonium bismuth citrate were too high (0.22 μM~2.2 mM, lanes 1–5), too low (2.2 pM, lane 10), or no ammonium bismuth citrate (lane PC), little or even no products amplified. It indicates that high concentrations have an inhibitory effect compared to PC, and a too low concentration can not effectively enhance the reaction. In addition, ammonium bismuth citrate has the best enhancement at a concentration of 0.22 nM (lane 8) on PCR, with up to approximately two times the amount of target product compared to the positive control that only has 3% DMSO and 5% glycerol added (Figure 2). 

Experiments on amplification of GNAS1 promoters by bismuth subcarbonate found that it can effectively enhance the amplification by addition of 0.2 mM (lane 2) bismuth subcarbonate. Similar to ammonium bismuth citrate, high concentrations (2.0 mM, lane 1) of bismuth subcarbonate have an inhibitory effect on PCR amplification, and too low concentrations (2.0 pM~22 μM, lanes 3–10) cannot effectively enhance the reaction (Figure 3a,c). In addition, to identify the strongest enhancing effect of bismuth subcarbonate on PCR amplification, the concentration of bismuth subcarbonate was refined in a two-fold gradient between 0.02 mM and 2 mM. It can be seen that the concentration of bismuth subcarbonate between 0.06 mM and 0.5 mM has a strong enhancing effect on the amplification reaction. As the concentration decreases, the amplification product increases until the concentration is down to 0.1 mM. The enhancing effect is the best compared with that of PC. Its target products are up to about four times. Then, the products reduced as the bismuth subcarbonate concentration decreased until the NOD was 0 (Figure 3b,d).

#### 2.3.2. Amplification of the APOE Gene

To test the ability of bismuth-based compounds to enhance GC-rich PCR, we chose the apolipoprotein E-APOE as a target gene [35]. As seen in Figure 4a,c, treating 0.22 μM~22 μM (lanes 3–5) ammonium bismuth citrate can also effectively enhance the amplification of the APOE gene, especially at 22 μM. Furthermore, 0.22 mM ammonium bismuth citrate completely inhibits reaction amplification. To detect the enhancing effect of ammonium bismuth citrate on PCR amplification between 22 μM and 0.22 mM, its concentration was refined by a two-fold gradient dilution. The results showed that the enhancing effect of ammonium bismuth citrate at 0.022 mM was still the best, and its target products are about 1.2-times higher than PC from NOD (Figure 4b,d). 

Through amplifying the APOE gene by bismuth subcarbonate, it was found that 20 nM~20 μM (lanes 3–6) bismuth subcarbonate could effectively enhance the amplification. There was a peak enhancement at 20 μM (lane 3) bismuth subcarbonate, which was approximately 1.4-times more target products than those by PC (Figure 5). Thus, bismuth subcarbonate in the appropriate concentration range is able to efficiently enhance the amplification of another GC-rich APOE gene by Ex Taq DNA polymerase.

#### 2.3.3. Applicability with Different Enzymes

To determine whether the reaction enhanced by bismuth-based compounds is suitable for the amplification of other common polymerases in PCR, we further analyzed the amplification of the GNAS1 promoter and APOE genes by six commonly used Taq DNA polymerases. Appendix A lists the six different polymerases, Vazyme Taq, BBI Taq, NEB Taq, Takara rTaq, Genstar Taq, and Toyobo rTaq DNA polymerases, along with the corresponding buffer in PCR system.

For the reaction system to amplify the GNAS1 promoter, as shown in Appendix A, the use of Vazyme Taq, BBI Taq, Takara rTaq, or Toyobo rTaq DNA polymerase can achieve effective amplification by adding only the additives-3% DMSO and 5% glycerol. The use of Vazyme Taq and Takara rTaq DNA polymerases amplified more target products with bismuth subcarbonate than with the addition of additives alone. The results indicate that bismuth subcarbonate-enhanced PCR is well suited to the systems in which these two enzymes are located. This is possibly attributed to the similar environments and the same pH. However, NEB Taq and Genstar Taq DNA polymerase could not amplify the product as their buffer solutions had a pH of 8.3, while the buffers in other enzyme systems had a pH higher than 8.3, indicating that solutions with different pH values would have a certain effect on PCR.

For the reaction system to amplify the APOE gene, as shown in Appendix A, the use of Vazyme Taq, NEB Taq, Takara rTaq, and Genstar Taq DNA polymerase can have weak target bands with the additives added, indicating that the products amplified very little. When treated with ammonium bismuth citrate and bismuth subcarbonate, the reaction can be enhanced using Vazyme Taq, Takara rTaq, Genstar Taq, and Toyobo rTaq DNA polymerase. Moreover, the PCR using NEB Taq DNA polymerase can only be enhanced with bismuth subcarbonate. In short, in the systems using Vazyme Taq, Takara rTaq, and Genstar Taq DNA polymerase, ammonium bismuth citrate-enhanced PCR can be well adapted to the environment of these three enzymes. In the system using Toyobo rTaq DNA polymerase, bismuth subcarbonate-enhanced PCR is also well adapted to the environment in which the enzyme is located.

In summary, ammonium bismuth citrate and bismuth subcarbonate-enhanced PCR amplification can be adapted to different polymerase reaction systems in similar environments.

### 2.4. Mechanism of Materials with PCR Components

#### 2.4.1. Bismuth-Based Materials Decrease the Melting Temperature (T_m_) of Primers

To explore the effect of bismuth-based compounds on primers, we observed the variation of the fluorescence signal with temperature by labelling the reporting fluorophore Cy5 and quenching group BHQ3 at the 5’ end (or 3’ end) of the forward primer (or reverse primer) and its complementary strand, respectively. The DNA duplex was unwound with the temperature gradually increasing from room temperature to 95 °C, so that the reporter fluorophore and the quenching group were separated. Thus, the fluorescence monitoring system could receive the fluorescence signal emitted by the reporter fluorophore. It was then slowly cooled to 25 °C, the complementary double strands were paired with each other, and the fluorescence signal emitted by the reporter was absorbed by the quenching group, making the fluorescence signal undetectable (Figure 6).

We tested the T_m_ values of primers (ASPro4se-FP, ASPro4as-RP) and their perfectly complementary strands (ASPro4se-FP-C, ASPro4as-RP-C) in the presence or absence of ammonium bismuth citrate or bismuth subcarbonate and compared with those with addition of additives. In the excitation, reactive oxygen species (ROS) are produced, which are very active and will attack the double bond, causing the internal double bond of the Cy5 dye to break, resulting in a decrease of the fluorescence signal [36]. So, we took the fluorescence signal of the oligonucleotide chain labelled with the Cy5 dye as a control, and the fluorescence signal was the ratio of the control. As shown in Figure 7a,c, the two strands are paired with each other at room temperature, and the quenching group quenches the signal emitted by the fluorescent reporter, basically producing no fluorescence signal. Then the molecular thermal movement intensifies with the temperature rising from 25 °C to the curve inflection point, and the double strand is opened with the enhancing fluorescence signal. Next, it is slowly warmed up until 95 °C, the double strand is unwound, and the fluorescence signal reaches the maximum. The experimental group with additives or materials had a higher fluorescence signal than the control. This is possibly because the additives or materials selectively adsorbed exposed single strands (ssDNA), which had different binding abilities to ssDNA and double-stranded DNA (ds DNA) [37,38,39]. Thus, the complementary paired dsDNA was reduced to enhance the fluorescence signal. In addition, ssDNA is much softer than dsDNA, with the persistence lengths of ssDNA and dsDNA as 1 and 50 nm, respectively, Bismuth-based compounds are much larger than ssDNA and cause ssDNA to wrap at the surface of bismuth-based compounds [40]. It can be observed that both ammonium bismuth citrate and bismuth subcarbonate have a higher fluorescence signal than addition of 3% DMSO and 5% glycerol, indicating that the materials have adsorption capacities for ssDNA. For the forward primers, the T_m_ value was 65 °C for the control and 62 °C for the experimental group. For the reverse primer, the T_m_ value was 64 °C for the control and 60 °C for the experimental group. Further, the presence of additives and bismuth-based compounds with the forward and reverse primer and their perfectly complementary strands resulted in a reduction in T_m_ values of 3.0 and 4.0 °C, respectively (Figure 7b,d). It may be attributed to the fact that DMSO [41,42] and glycerol [43] can bind to the ssDNA region when annealed, preventing rehybridization, and reducing the T_m_ values of complementary and mismatched primers to increase the differences of T_m_ values between them, which then affects the annealing temperature required for PCR to maintain high specificity. The difference in T_m_ values between forward and reverse primers may be due to the difference in affinity between ammonium bismuth citrate or bismuth subcarbonate and the four nucleotides.

We also tested the T_m_ values of primers (APOE-FP, APOE-RP) and their perfectly complementary strands (APOE-FP-C, APOE-RP-C) in the presence or absence of ammonium bismuth citrate or bismuth subcarbonate and compared with those with addition of additives. As shown in Figure 8a,c, similar to the forward and reverse primers for amplifying the GNAS1 promoter described above, the fluorescence signal of the experimental group with additives or materials was significantly higher than that of the control. Further, it can be observed that the fluorescence signal of ammonium bismuth citrate is much higher than that of bismuth subcarbonate and the presence of 3% DMSO and 5% glycerol, indicating that ammonium bismuth citrate has a better affinity for the bases present in APOE-FP (Figure 8a,c). The presence of 0.022 mM ammonium bismuth citrate or 0.02 mM bismuth subcarbonate both caused 3.0 °C decreases of both T_m_ values of the forward and reversed primer with their perfect complementary strands compared to those of control (Figure 8b,d).

#### 2.4.2. Bismuth-Based Materials Adsorb Polymerases and Regulate the Number of Active Polymerases in PCR 

Non-toxic bismuth salts are usually used in anti-ulcer medications to protect against nephrotoxicity from anticancer drugs [44,45]. These effects are closely associated with the protein. A typical PCR contains a variety of components, including dNTP, primers, template DNA, dsDNA products, DNA polymerases, and some additives. In situ observation and quantification of multi-component competitive adsorption on surfaces are quite tricky. Some studies show that bismuth salts can induce metallothionein (MT), a metal-binding protein that lacks a formal secondary structure [46]. He et al. demonstrated that Bi (III) binds to HpFur protein, specifically at the physiologically important S1 site, which further leads to protein oligomerization and loss of DNA binding capability [47]. Cheek et al. determined that Bi (III) and L-cysteine (Cys) form soluble complexes at both pH 1.0 and pH 7.4 [48]. Bi (III) always binds to proteins in some way. Here, the adsorption of polymerase on bismuth-based materials was confirmed by using proteins at five different isoelectric points, BSA, TB, 𝛾-Gl, Lys, and casein, as competitive sorbents to recover the inhibition of PCR by excess bismuth-based materials.

As shown in Figure 9a, under the inhibition with high concentrations of ammonium bismuth citrate, it was found that 1 μM and 10 μM BSA could partially restore the inhibitory effect of excessive ammonium bismuth citrate. BSA’s isoelectric point (pI) is 4.7, which is significantly lower than Taq (pI 6.0), whereas the pH of the reaction was about 8.9. Therefore, BSA will have more negative charge than Taq, making it have a stronger electrostatic attraction with positively charged Bi (III) and amino groups of ammonium bismuth citrate. Further, a relatively low concentration of BSA (1 μM) can effectively adsorb to the surface of ammonium bismuth citrate. However, it is well known that BSA can improve the efficiency of PCR [49]. High concentrations of BSA in PCR systems may have unknown effects on PCR, making PCR systems more complex. From this point of view, BSA may not be the most ideal PCR additive for mechanistic studies to regulate polymerase adsorption. Therefore, we select casein (pI 4.6) that has an isoelectric point similar to BSA. We find that it can also partially restore the inhibitory effect of excess ammonium bismuth citrate, indicating that casein can effectively adsorb to the surface of ammonium bismuth citrate (Figure 9b). In addition, we also studied three other relatively high isoelectric proteases, namely TB (pI 7.0), 𝛾-Gl (pI 6.85), and Lys (pI 11.35). They all had higher isoelectric points than Taq (pI 6.0), while the reaction had a pH of about 8.9. Taq will carry more negative charges than TB and 𝛾-Gl, resulting in a stronger electrostatic attraction between positively charged ammonium bismuth citrate and Taq, reducing the concentration of Taq enzyme in the system and not restoring the inhibitory effect of ammonium bismuth citrate on PCR. Lys is positively charged in a system at pH 8.9, allowing it to have a robust electrical repulsion with the equally positively charged ammonium bismuth citrate, and cannot compete with Taq enzyme for adsorption to the surface of ammonium bismuth citrate. Therefore, it cannot adjust the amount of polymerase activity to restore PCR (Figure 9a).

In addition, we also investigated the effect of bismuth subcarbonate at high concentration on polymerase by incomplete inhibition of it. In contrast to positively charged ammonium bismuth citrate, bismuth subcarbonate is negatively charged in PCR systems. As shown in Figure 9c,d, under the low isoelectric point BSA and casein, it was found that the higher electronegativity of 1 μM BSA and casein had stronger electrical repulsion between negatively charged bismuth subcarbonate. So, it could not be effectively adsorbed to bismuth subcarbonate. For the other three relatively high isoelectric points of protease-TB (pI 7.0), 𝛾-Gl (pI 6.85), and Lys (pI 11.35), their isoelectric points were all higher than Taq (pI 6.0), while the reaction had a pH of about 8.9. Taq will have more negative charges than TB and 𝛾-Gl and have stronger electrical repulsion with bismuth subcarbonate, which may be inefficiently adsorbed on bismuth subcarbonate, resulting in complete inhibition of PCR amplification. Lys is positively charged in a pH 8.9 system, and there is a strong electrostatic attraction between Lys and bismuth subcarbonate, which can be adsorbed to the surface of bismuth subcarbonate and also generates electrostatic attraction with negatively charged DNA, resulting in ineffective amplification.

Therefore, proteins with different isoelectric points can regulate the amount of polymerase activity by competitive adsorption with Ex Taq DNA polymerase in the PCR system, thereby restoring the inhibition of PCR amplification by high concentrations of bismuth-based materials. However, for different materials, the ultimate effect on PCR may vary under specific PCR conditions and should be evaluated on a case-by-case basis.

#### 2.4.3. Bismuth-Based Materials Promote the Dissociation of the Product

Lou et al. demonstrated that PCR efficiency can be improved by accelerating the dissociation of PCR products during the denaturation step assisted by Au NPs [29]. Inspired by this study, we amplified the target product by RT-qPCR and then measured the amount of dsDNA products as a function of heating temperature that is the melting curve of the product to study the effects of ammonium bismuth citrate and bismuth subcarbonate on the degree of product dissociation. Here, we used SYBR Green I fluorescent dye, which can bind non-specifically to ssDNA and emit weak fluorescence in the free state. Once it was embedded in dsDNA, the fluorescence signal was greatly enhanced. So, it can be used for real-time detection of ssDNA amplification products [50]. Appendix A shows the schematic diagram of the formation principle of the melting curve. After PCR amplification, the DNA is gradually dissolved with the increasing temperature, the SYBR Green I embedded in the dsDNA is also shed, and the fluorescence signal is weakened. When T_m_ is reached, the dsDNA is rapidly dissolved to half, and the fluorescence signal drops sharply. Subsequently, the temperature rises until the dsDNA is completely dissolved and the fluorescence signal is reduced to a minimum. Therefore, real-time detection of the fluorescence signal in this process can obtain the original fluorescence signal intensity data as a temperature function.

Figure 10a shows the amplification curve of the GNAS1 promoter. The Cq value of the 3% DMSO and 5% glycerol group was 30.91. Further, the Cq values of ammonium bismuth citrate and bismuth subcarbonate groups were 30.64 and 30.50, respectively. Figure 10b shows the melting curves of the GNAS1 promoter. The T_m_ value of the product (283 bp) was about 91 °C. Further, after adding ammonium bismuth citrate and bismuth subcarbonate, it was reduced to 90.5 °C by 0.5 °C. The dissociation percentage of the product is defined as (F_0_ − F)/F_0_, where F_0_ and F are the fluorescence intensity (RFU) at 87 °C and the set measurement temperature (93 °C). We found that the dissociation percentages of the products without and with 0.22 nM ammonium bismuth citrate and 0.1 mM bismuth subcarbonate were 91.8%, 92.8%, and 92.7%, respectively. In addition, we also confirmed by gel electrophoresis that ammonium bismuth citrate (lane 2) and bismuth subcarbonate (lane 3) can promote product dissociation and increase the yield of the products (Figure 10c). The data clearly show that surface interactions enhance the product’s dissociation and prevent its rehybridization like SSB, enhancing reaction specificity.

Figure 11 shows the amplification of the APOE gene. The Cq value of the 3% DMSO and 5% glycerol group was 22.49. Further, the Cq values of ammonium bismuth citrate and bismuth subcarbonate groups were 22.30 and 22.26, respectively. The T_m_ value of the product (322 bp) was about 92.5 °C; after adding ammonium bismuth citrate and bismuth subcarbonate, the T_m_ value of the product was reduced to 92 °C and reduced by 0.5 °C (Figure 11b). Here, F is set as 95 °C. We found that the dissociation percentages of PCR products without and with 0.022 mM ammonium bismuth citrate and 0.02 mM bismuth subcarbonate were 94.6%, 95.0%, and 95.1%, respectively. In addition, we also confirmed through gel electrophoresis experiments that ammonium bismuth citrate (lane 2) and bismuth subcarbonate (lane 3) can promote product dissociation and increase the yield of amplification products (Figure 11c), which is similar to the effect of amplifying GNAS1 promoter.

## 3. Experiment

### 3.1. Materials and Apparatus

Ammonium bismuth citrate and bismuth subcarbonate were purchased from Adamas Reagent, Ltd. DMSO and glycerol from Sinopharm Chemical Reagentco., Ltd. The PCR reagents including dNTPs, 10×PCR Buffer (Mg^2+^ free), MgCl_2_, and Ex Taq DNA polymerases were obtained from Takara. Other Taq DNA polymerases were purchased from Vazyme, Sangon, NEB, Takara, Genstar, and Toyobo. Human genomic DNA (g-DNA) was extracted from RD and U87-MG cells. All oligonucleotide sequences used in this article are shown in Table 2. The primers were all synthesized by BioSune Biotech (Shanghai, China) Co., Ltd.

Conventional PCR was carried out using ABI Proflex 9 (Applied biosystems by life technologies), the qRT-PCR was carried out using LightCycle 96 (Roche), and the melt curve experiments were performed using a CFX96 (Bio-Rad) fluorescence PCR instrument. All experiments were repeated twice to three times to confirm that the results are true and valid.

### 3.2. PCR Systems

#### 3.2.1. Preparation of Working Solution 

First, we prepared the solvents for use in the materials, DMSO and glycerol (together with enzymes, it was determined by a three-factor two-level orthogonal table shown in Table 1). After weighing, 5 mg ammonium bismuth citrate and bismuth subcarbonate were dissolved in 1 mL prepared solvent. Then the solutions were ultrasonic until stable suspensions were formed. Finally, we diluted the two materials in a 5 × 10^−9^ mg/mL gradient. They were stored at room temperature.

#### 3.2.2. Conventional PCR Amplification

(1)Materials used in amplification reactions

A quantity of 10 μL of prepared ammonium bismuth citrate and bismuth subcarbonate (5 mg/mL–5 × 10^−9^ mg/mL) were taken (1 mg/mL–1 pg/mL, final concentration in PCR system, the same below) for amplification of GNAS1 and APOE, two GC-rich systems. Each reaction was performed with a final volume of 50 μL, which commonly contained 0.4 μM GNAS1 or 0.2 μM APOE in each primer, 0.2 mM each dNTP (dATP, dCTP, dTTP, and dGTP), 100 ng (GNAS1) or 50 ng (APOE) of Homo sapiens g-DNA, 1.25 U Ex Taq DNA polymerase, 1.5 mM Mg^2+^, and 1× Ex Taq Buffer.

For amplifying the GNAS1 promoter using Ex Taq DNA polymerase, a “touchdown” PCR program followed predenaturation at 94 °C for 5 min, consisting of 25 cycles of amplification: 1 min at 94 °C, 30 s at annealing temperature, and 40 s at 72 °C. The annealing temperature decreased by 0.5 °C every cycle from 72 to 60 °C. Then, a standard PCR program was performed for 20 cycles (1 min at 94 °C, 30 s at 60 °C, and 40 s at 72 °C). 

When amplifying the APOE gene, predenaturation at 96 °C for 5 min was followed by 30 cycles of 45 s at 94 °C, 30 s at 60 °C, and 20 s at 72 °C. Finally, the overall extension stage reacted at 72 °C for 5 min.

(2)Different Taq DNA polymerases in amplification reactions

Different brands of Taq polymerase and corresponding buffer solutions, including Vazyme, Sangon, NEB, Takara, Genstar, and Toyobo, were applied to the above-optimized system. 

(3)The effect of the materials on polymerase

Six enzymes with different isoelectric points, BSA, TB (bovine thrombin), 𝛾-Gl (𝛾-globulin), Lys (lysozyme), and casein, were used in the adsorption experiment of material for protease. Five different concentrations of proteases were prepared with 0.9% NaCl solution to 0.1 μM, 1 μM, and 10 μM for backup. The system-added template was used as the negative control; 3% DMSO and 5% glycerol were added as the positive control; another group was added to the material suppression group, and then six different isoelectric points of protease solutions with three concentration gradients were added to the solution of the material suppression group. Ammonium bismuth citrate and bismuth subcarbonate were designed with two sets of experiments, and other conditions were the same.

### 3.3. Gel Electrophoresis

The experiment required the preparation of agarose gels with a concentration of 2%, and different volumes of gels were prepared according to the number of tubes of different experimental reaction solutions. The electrophoresis channel was set at a voltage of 130 V and 400 mA current flow for 35 min. Electrophoresis was performed until a blue band of electrophoresis appeared around the middle of the gel. The isolated DNA fragments with the Tanon 2500 digital gel image processing system(GIS) were observed and analyzed.

### 3.4. Fluorescence Measurement of Melting Temperature (T_m_) of Primers

Each 50 μL sample contained 1 × Ex Taq PCR Buffer, forward primer (ASPro4se-FP) and ASPro4se-FP-C or reverse primer (ASPro4se-RP) and ASPro4se-RP-C, followed by the addition of ammonium bismuth citrate (0.22 nM) and bismuth subcarbonate (0.1 mM) for GNAS1 promoter. For the APOE gene, forward primer (APOE-FP) and APOE-FP-C or reverse primer (APOE-RP) and APOE-RP-C were present, followed by addition of ammonium bismuth citrate (0.022 mM) and bismuth subcarbonate (0.02 mM). 

The melting procedure was performed for 2 min at 95 °C, starting from 25 °C, raising by 0.5 °C/min to 95 °C, and continuously collecting fluorescence signals.

### 3.5. Fluorescence Measurement of Dissociation Percentage of PCR Products

The 50 μL reaction mixtures contained 1.25U Ex Taq DNA polymerase, 0.2 mM each dNTPs, 1 × SYBR Green I, Homo sapiens g-DNA (100 ng), 1 × Ex Taq Buffer, primers (0.4 μM ASPro4se-FP, ASPro4se-RP or 0.2 μM APOE-FP, APOE-RP), and 1.5 mM Mg^2+^ (GNAS1) or 2 mM Mg^2+^ (APOE). Materials were the last ones added to the tubes. PCR cycling conditions were the same as those for conventional PCR for GNAS1 and APOE. The melting program was set as follows: first 60 s (4.40 °C/s) at 95 °C, 60 s (2.20 °C/s) at 40 °C, 1 s (2.20 °C/s) at 65 °C, and finally 1 s at 97 °C, from 65 °C to 97 °C at a rate of 0.07 °C/s, and fluorescence signal data were continuously collected.

## 4. Conclusions

We investigated the role of two different Bi-based materials, ammonium bismuth citrate and bismuth subcarbonate, in PCR and the interaction mechanism between the materials and the main components (primers, polymerases, and products) of the PCR system. 

Through PCR and gel electrophoresis experiments, it was found that both ammonium bismuth citrate and bismuth subcarbonate can effectively increase the reaction yield and enhance the specificity of the reaction within a certain concentration range, which opens up a new way for their application in in vitro amplification of nucleic acid chains and other fields. Moreover, the mechanism studies showed that the two Bi-based materials mainly reduced the T_m_ value by adsorbing primers, thereby reducing the reaction’s annealing temperature and improving the reaction’s specificity. At the same time, the activity of polymerase in the reaction solution is regulated to enhance the reaction efficiency through the adsorption of enzymes. It is also proved that the material can also promote the dissociation of the product, accelerate the reaction process, and increase the yield of the target product. The effects of ammonium bismuth citrate and bismuth subcarbonate on PCR are based on surface interactions with PCR components. In the reaction process, these components are dynamically adsorbed and dissociated from the surface of bismuth-based materials. Therefore, the ultimate effect of ammonium bismuth citrate and bismuth subcarbonate on PCR may vary from case to case, and specific applications may require careful optimization. 

In summary, the two bismuth compounds assisted by DMSO and glycerol can not only effectively enhance the specificity of the reaction but also increase the product yield in GC-rich PCR amplification. Bismuth-based materials have great potential for application in the field of PCR amplification and can be extended to rapid amplification of nucleic acids in vitro by different methods or instruments. 

## Figures and Tables

**Figure 1 molecules-28-04515-f001:**
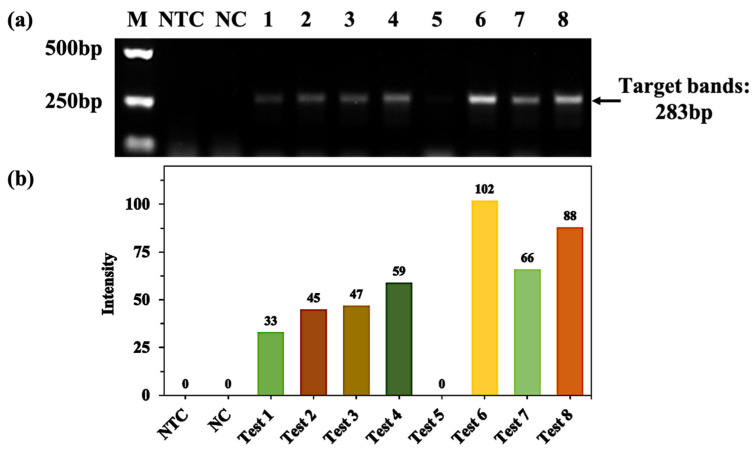
(**a**) PCR amplification of Ex Taq DNA polymerase-mediated GNAS1 promoter was optimized using different ratios of enzymes, DMSO, and glycerol with 1.5 mM Mg^2+^. Lane M: 100 bp DNA ladder (DL2000 Plus, Vazyme; DL1001, Generay); lane NTC: no template control; lane NC: negative control with templates; lanes 1–8: optimal amplification of combinations of enzymes, DMSO, and glycerol in different proportions, as detailed in Table 1; (**b**) the fluorescent intensity corresponds to the target band of the gel in figure (**a**).

**Figure 2 molecules-28-04515-f002:**
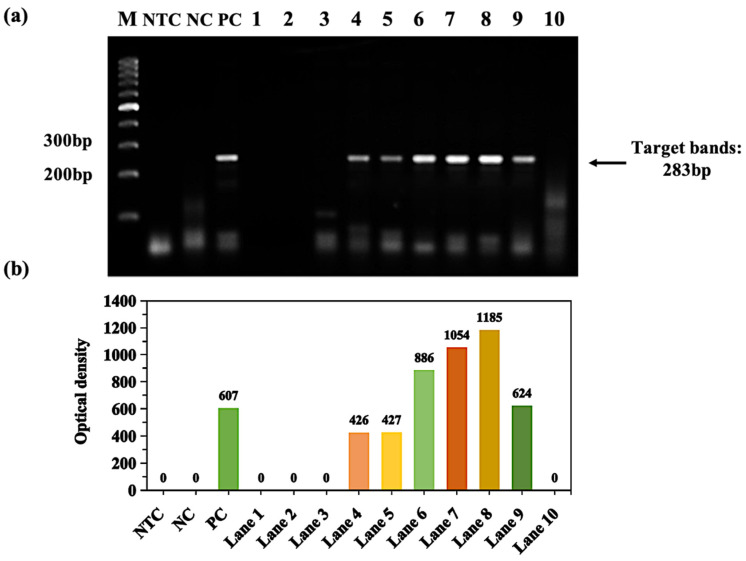
Amplification of GNAS1 promoter enhancement by ammonium bismuth citrate in GC-rich PCR. (**a**) Gradient dilution of ammonium bismuth citrate from 2.2 mM to 2.2 pM (lanes 1–10); lane M: 100 bp DNA ladder (DL1002, Generay); lane NTC: no template control; lane NC: negative control with templates; lane PC: 3% DMSO and 5% glycerol; (**b**) the NOD corresponds to the target band of the gel in figure (**a**).

**Figure 3 molecules-28-04515-f003:**
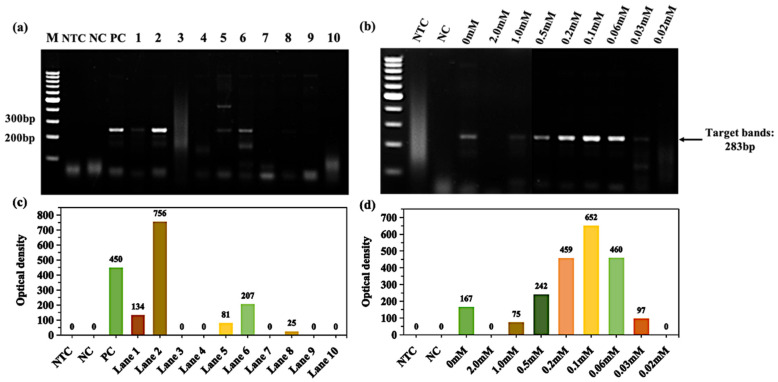
Enhancement of GNAS1 promoter amplification by bismuth subcarbonate in GC-rich PCR. (**a**) Bismuth subcarbonate is diluted in a gradient from 2.0 mM to 2.0 pM (lanes 1–10); (**b**) Bismuth subcarbonate is diluted at a double gradient between 2.0 mM and 0.02 mM, and the concentration of materials used in each lane is marked. Lane M: 100 bp DNA ladder (DL1002, Generay); lane NTC: no template control; lane NC: negative control with templates; lane PC: 3% DMSO and 5% glycerol; the NOD (**c**,**d**) corresponds to the target bands in the gel of figure (**a**,**b**), respectively.

**Figure 4 molecules-28-04515-f004:**
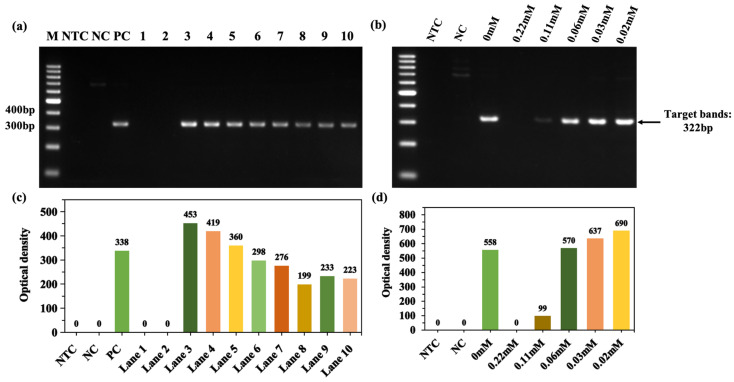
Ammonium bismuth citrate amplifies the enhancement of APOE gene in GC-rich PCR. (**a**) Ammonium bismuth citrate was diluted from 2.2 mM to 2.2 pM (lanes 1–10); (**b**) ammonium bismuth citrate was diluted in a double gradient between 0.22 mM and 0.02 mM, and the concentration of materials used in each lane is marked. Lane M: 100 bp DNA ladder (DL1002, Generay); lane NTC: no template control; lane NC: negative control with templates; lane PC: 3% DMSO and 5% glycerol; the NOD (**c**,**d**) corresponds to the target bands in the gel of figure (**a**,**b**), respectively.

**Figure 5 molecules-28-04515-f005:**
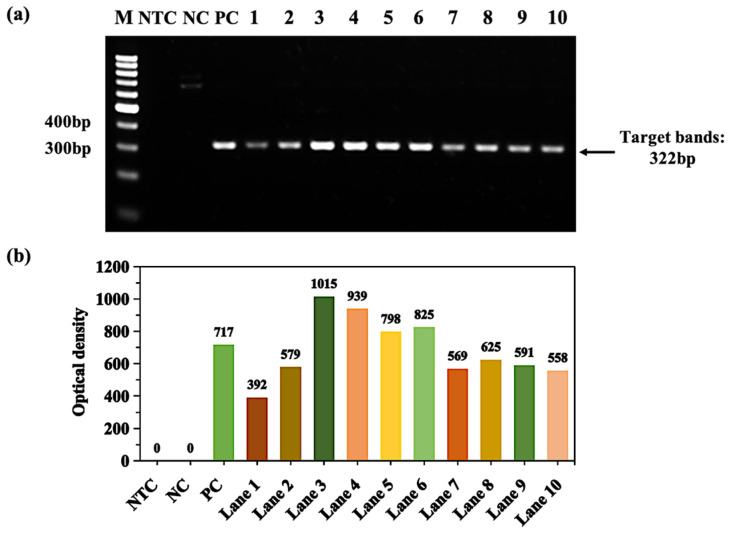
Enhancement of APOE gene amplification by bismuth subcarbonate in GC-rich PCR. (**a**) Bismuth subcarbonate gradient dilution from 2.0 mM to 2.0 pM (lanes 1–10); lane M: 100 bp DNA ladder (DL1002, Generay); lane NTC: no template control; lane NC: negative control with templates; lane PC: 3% DMSO and 5% glycerol; (**b**) the NOD corresponds to the target band in the gel in figure (**a**).

**Figure 6 molecules-28-04515-f006:**
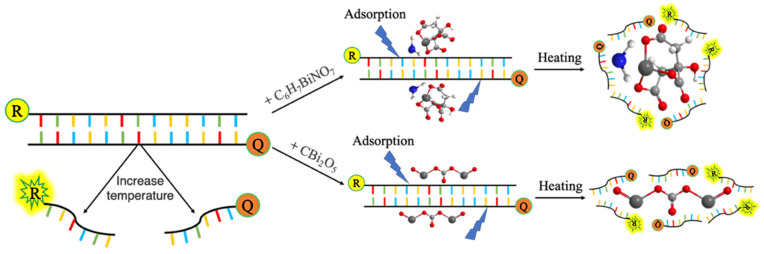
The mechanism for the adsorption of oligonucleotide chains by ammonium bismuth citrate and bismuth subcarbonate.

**Figure 7 molecules-28-04515-f007:**
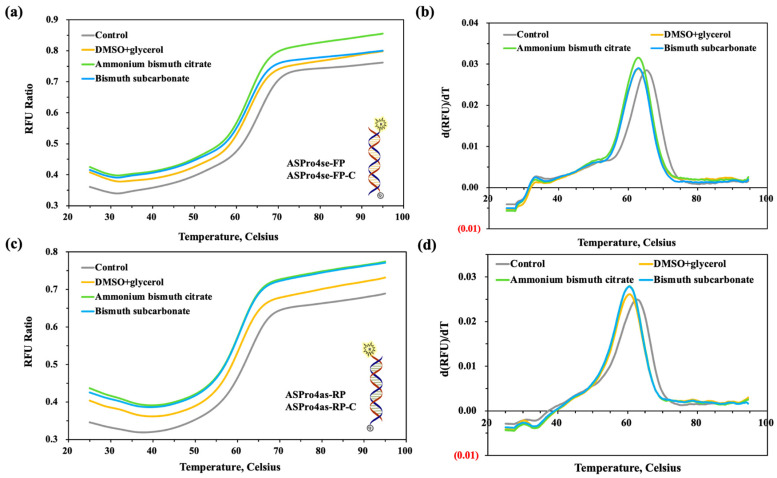
The pattern of fluorescence signal changes of (**a**,**c**) primer melting curves and (**b**,**d**) primer melting peaks of oligonucleotide chains (ASPro4se-FP or ASPro4se-RP) adsorbed by ammonium bismuth citrate or bismuth subcarbonate, corresponding to forward or reverse primers amplified by GNAS1 promoters in PCR. Control group: 1 × Ex Taq buffer; experimental group: 3% DMSO + 5% glycerol, 0.22 nM ammonium bismuth citrate, 0.1 mM bismuth subcarbonate.

**Figure 8 molecules-28-04515-f008:**
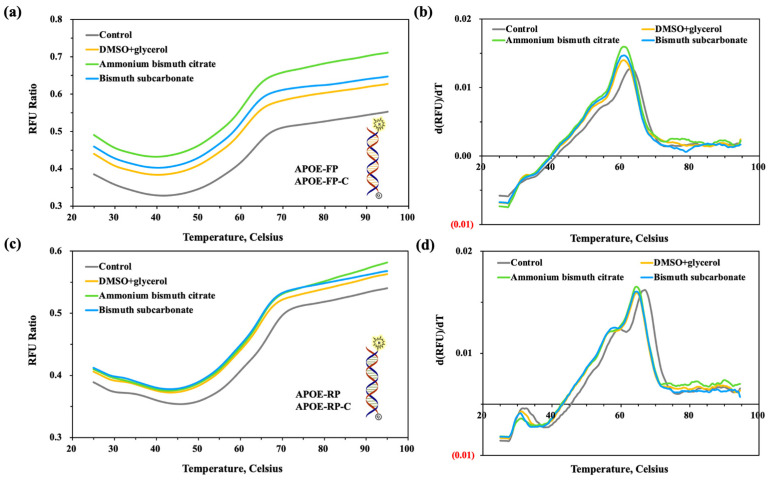
Diagram of the (**a**,**c**) primer melting curve and (**b**,**d**) primer melting peak of an oligonucleotide chain (ASPro4se-FP or ASPro4se-RP) adsorbed by ammonium bismuth citrate or bismuth subcarbonate, corresponding to the forward primer or reverse primer amplifying APOE gene in PCR and its complementary strand (ASPro4se-FP-C or ASPro4se-RP-C). Control group: 1 × Ex Taq buffer; experimental group: 3% DMSO + 5% glycerol, 0.022 mM ammonium bismuth citrate, 0.02 mM bismuth subcarbonate.

**Figure 9 molecules-28-04515-f009:**
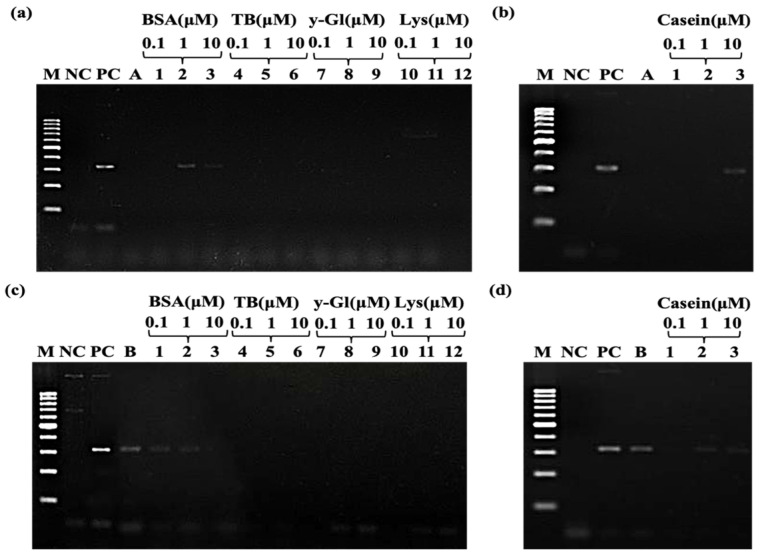
PCR amplification of APOE gene regulated by polymerase-based materials. Recovery of the inhibitory effect of (**a**) ammonium bismuth citrate and (**c**) bismuth subcarbonate on PCR by BSA, TB, 𝛾-Gl, and Lys; restoration of the inhibitory effect of (**b**) ammonium bismuth citrate and (**d**) bismuth subcarbonate on PCR by casein; lane M: 100 bp DNA ladder (DL1002, Generay); lane NC: negative control with template; lane PC: positive control using a mix of 3 % DMSO and 5 % glycerol; lane A/B: containing ammonium bismuth citrate (0.1 mg/mL)/bismuth subcarbonate (1 mg/mL); lanes 1–12: addition of different concentrations of (a,c) BSA (lanes 1–3), TB (lanes 4–6), 𝛾-Gl (lanes 7–9), and Lys (lanes 10–12) and (b,d) casein (lanes 1–3), other conditions are the same as lanes A/B.

**Figure 10 molecules-28-04515-f010:**
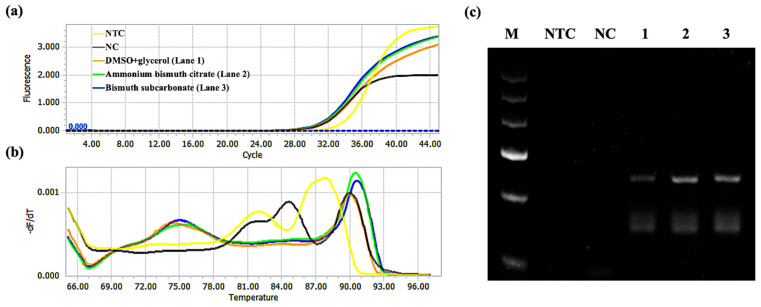
RT-qPCR and gel electrophoresis for the effects of ammonium bismuth citrate and bismuth subcarbonate on the degree of dissociation of GNAS1 promoter amplification products. (**a**) amplification curve; (**b**) product melting curve; (**c**) gel electrophoresis of products, lanes 1–3: 3% DMSO and 5% glycerol, 0.22 nM ammonium bismuth citrate, and 0.1 mM bismuth subcarbonate, lane M: 100 bp DNA ladder (DL1001, Generay); lane NTC: no template control; lane NC: negative control with templates.

**Figure 11 molecules-28-04515-f011:**
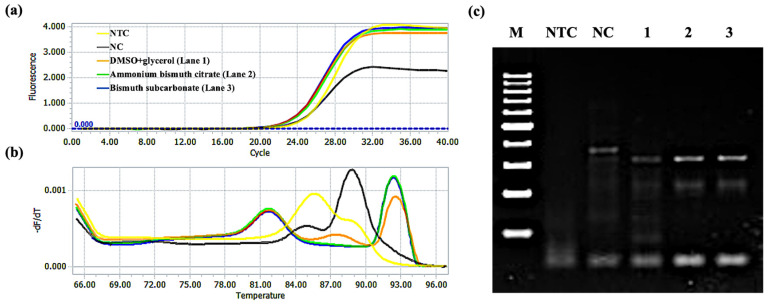
RT-qPCR and gel electrophoresis detected the effects of ammonium bismuth citrate and bismuth subcarbonate on the degree of dissociation of APOE gene amplification products. (**a**) amplification curve; (**b**) product melting curve; (**c**) gel electrophoresis of products, lanes 1–3: 3% DMSO and 5% glycerol, 0.022 mM ammonium bismuth citrate, and 0.02 mM bismuth subcarbonate, lane M: 100 bp DNA ladder (DL1002, Generay); lane NTC: no template control; lane NC: negative control with templates.

**Table 1 molecules-28-04515-t001:** Three-factor two-level orthogonal table.

Test	Enzyme	DMSO	Glycerol
Test 1	1.25U	3%	5%
Test 2	1.25U	3%	10%
Test 3	1.25U	6%	5%
Test 4	1.25U	6%	10%
Test 5	2.50U	3%	5%
Test 6	2.50U	3%	10%
Test 7	2.50U	6%	5%
Test 8	2.50U	6%	10%

**Table 2 molecules-28-04515-t002:** List of all the oligonucleotide sequences used in this work.

Name	Sequence (5′-3′)	Description
ASPro4se-FP	GAGCGTTGGCGTCGTGC (17 bp)	Forward primer for GNAS1 promoter
	GAGCGTTGGCGTCGTGC-Cy5-3’	
ASPro4as-RP	GAGGAGGAGGGCCGAGGA (18 bp)	Reverse primer for GNAS1 promoter
	5’-Cy5-GAGGAGGAGGGCCGAGGA	
ASPro4se-FP-C	GCACGACGCCAACGCTC-BHQ3-3’	Complementary to ASPro4se-FP
ASPro4as-RP-C	5’-BHQ3-TCCTCGGCCCTCCTCCTC	Complementary to ASPro4as-RP
APOE-FP	CCCGGTGGCGGAGGAGACG (19 bp)	Forward primer for APOE gene
	CCCGGTGGCGGAGGAGACG-Cy5-3’	
APOE-RP	GTCGCGGCCCTGTTCCACCAG (21 bp)	Reverse primer for APOE gene
	GTCGCGGCCCTGTTCCACCAG-BHQ3-3’	
APOE-FP-C	CGTCTCCTCCGCCACCGGG-BHQ3-3’	Complementary to APOE-FP
APOE-RP-C	CTGGTGGAACAGGGCCGCGAC-Cy5-3’	Complementary to APOE-RP

## Data Availability

Not applicable.

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
