# Peer review of "Enhancement Effects and Mechanism Studies of Two Bismuth-Based Materials Assisted by DMSO and Glycerol in GC-Rich PCR"

_molecules, 2023, doi:10.3390/molecules28114515_

Round 1

Reviewer 1 Report

In this research, the author has introduced two different Bi-based materials-ammonium bismuth citrate and bismuth subcarbonate in PCR to enhance the reaction efficiency and specificity. This study has suggested a new class of PCR enhancer with simple, low-cost, and efficient characteristics.  This is exciting research and suitable for the journal. The following comments may help the authors to improve the manuscript before acceptance.

-          Please check word spacing in the sentence.

-          Revise the manuscript again to avoid grammatical mistakes.

-          Make sure the texts in manuscript are in the same font.

Besides these comments about format and grammar, I have some questions regarding the methods and data as below:

1.     In Fig. 1, please add some quantitative data such as fluorescent intensity measured from gel results.

2.     In Section 3.1, (1) Optimization for amplifying GNAS1 promoters, the author choose test 1 for further experiment because of “the cost of the enzyme, the practicality of the enhancer combination, and the amplification effect of genes with high GC content”. However, based on the gel result in Fig. 1, the intensity of test 1 seems very poor as compared with others. Please discuss more about the reason to select it although its performance is not comparable to others.

3.     How do you measure optical density?

4.     Fig. 11 should be moved to Supplementary.

5.     In Fig. 5, Fig. 5d showed net optical densities correspond to the target bands in the gel of Fig. 5b. In Fig. 5b, I see the intensities of treatment 0.03 mM is brighter than treatment 0.11 mM; however, in Fig. 5d, the net optical densities of the two treatments are opposite. Can you explain?

-          Please check word spacing in the sentence.

-          Revise the manuscript again to avoid grammatical mistakes.

-          Make sure the texts in manuscript are in the same font.

Author Response

To Reviewer #1:

  1. In Fig. 1, please add some quantitative data such as fluorescent intensity measured from gel results.

Reply: We greatly appreciate your valuable suggestion and comments. We have added the fluorescent intensity measured from gel results by digital gel image processing system. (see page 6, Fig.1b)

  1. In Section 3.1, (1) Optimization for amplifying GNAS1 promoters, the author choose test 1 for further experiment because of “the cost of the enzyme, the practicality of the enhancer combination, and the amplification effect of genes with high GC content”. However, based on the gel result in Fig. 1, the intensity of test 1 seems very poor as compared with others. Please discuss more about the reason to select it although its performance is not comparable to others.

Reply: Thank you very much for focusing on this issue. Our main purpose in choosing test 1 is to make the control group with only DMSO and glycerol more obvious compared with the experimental group after adding the materials. We've discussed more reasons than the last version. (see page 5, section 3.1, paragraph 2)

  1. How do you measure optical density?

Reply: Thank you for your question. We use the Tanon 2500 digital gel image processing system(GIS) for data quantification, as the added description. (see page 4, section 2.3)

  1. 11 should be moved to Supplementary. 

Reply: According to your nice suggestion, we have moved Fig.11 as Fig.S7 of the supplementary. (see pages 16-17, section 3.4.3)

  1. In Fig. 5, Fig. 5d showed net optical densities correspond to the target bands in the gel of Fig. 5b. In Fig. 5b, I see the intensities of treatment 0.03 mM is brighter than treatment 0.11 mM; however, in Fig. 5d, the net optical densities of the two treatments are opposite. Can you explain?

Reply: We're sorry that the net optical density data was entered incorrectly. In Fig.5b (now Fig.4b), the experimental results showed that the intensities of treatment 0.03 mM is indeed brighter than treatment 0.11 mM; actually, in Fig.5d (now Fig.4d), the net optical densities of treatment 0.03 mM is higher than treatment 0.11 mM, and treatment 0.06 mM have also been corrected. (see page 10, section 3.3.2, Fig.5d is now Fig.4d)

Reviewer 2 Report

Review:

Enhancement effects and mechanism studies of two bismuth-based materials assisted by DMSO and glycerol in GC-rich PCR

In this study, two bismuth-based materials were used to optimize GC-rich PCR. It uses a series of methods to find answers to important questions. It makes use of a variety of precise methods that are suitable for achieving the objectives set. A large amount of data has been generated. These data are processed separately and in context. Ammonium bismuth citrate and bismuth subcarbonate were effective in enhancing PCR amplification of GC-rich promoter and encoding regions using Ex Taq DNA. A combination 3% DMSO and 5% glycerol were used as additives to obtaining the target amplicons. Moreover, Tm reduced. Every combination is tested in different concentrations. Do you know any other cheap materials in use with comprehensive optimization effects and simple preparation methods? How can You explain the differences between the amplification results of the promoter and the gene? Does bismuth complexes affect the further use of PCR products?

The graphs are sufficiently detailed and easy to expound. The text contains occasional typos, but it is nevertheless easy to read and uses scientific language appropriately. The literature used is fairly extensive. Add a “space” before “[“ and after “%” in every case. Latin names should be in Italics. A dot remained in line 199. Fig 2, Fig 11, and Table 3 take up more space in the manuscript than their content warrants. These are not informative enough so please consider removing them. In my opinion, Validation belongs to Methods. Fig 12-13 would be more unambiguous with the same marking of the lines of Fig 8-9.

The text contains occasional typos, but it is nevertheless easy to read and uses scientific language appropriately. 

Author Response

To Reviewer #2:

  1. Do you know any other cheap materials in use with comprehensive optimization effects and simple preparation methods?

Reply: According to the literature reviewed so far, it is searched that there are no simple and cheap materials that can not only improve the specificity of amplification reactions but also increase product yield in PCR. Nanomaterials have a comprehensive optimization effect, but most nanoparticles are relatively complex to prepare; Other biochemical additives such as betaine, dimethyl sulfoxide (DMSO), formamide, glycerol, non-ionic detergents, and bovine serum albumin (BSA) are not always effective in enhancing the reaction.

  1. How can you explain the differences between the amplification results of the promoter and the gene?

Reply: Thank you for asking the thoughtful question. We have thought about this before, and learned from consulting some literature that the difference in amplification results between promoters and genes may be due to affinity differences between the material and the four nucleotides.

  1. Do bismuth complexes affect the further use of PCR products?

Reply: Many thanks for your valuable comments. It depends on what type of further experiments, the material itself is not involved in the chemical reaction, and the properties of the bismuth complex itself do not change. After the PCR reaction, we found that the high concentration of bismuth complex would have a white precipitate at the bottom of the reaction tube. If there is any possible effect on subsequent product experiments, the supernatant can be taken by centrifugation to minimize the impact.

  1. Add a “space” before “[” and after “%” in every case. Latin names should be in Italics. A dot remained in line 199. Fig 2, Fig 11, and Table 3 take up more space in the manuscript than their content warrants. These are not informative enough so please consider removing them. In my opinion, Validation belongs to Methods. Fig 12-13 would be more unambiguous with the same marking of the lines of Fig 8-9.

Reply: Thank you for your careful correction. All formatting issues have been corrected. Fig. 2, 11, and Table 3 were moved as Fig. S4, Fig. S7, and Table S1 in the supplementary, and the discussions and results are briefly analyzed. We have mapped the line markers in Fig. 8-9 (now Fig. 7-8) to Fig. 12-13 (now Fig. 10-11). (see pages 6-7, 12-13, 15-17, section 3.2, 3.4.1, 3.4.3)

Reviewer 3 Report

To,

The Editor,

Molecules, MDPI,

Manuscript ID: molecules-2345293

Subject: Submission of comments of the manuscript in “Molecules"

Dear Editor Molecules, MDPI,

Thank you very much for the invitation to consider a potential reviewer for the manuscript (ID: molecules-2345293). My comments responses are furnished below as per each reviewer’s comments. 

In the reviewed manuscript, the authors' two bismuth-based materials that are inexpensive and readily available were used to optimize GC-rich PCR. The results demonstrated that ammonium bismuth citrate and bismuth subcarbonate were effective in enhancing PCR amplification of the GNAS1 promoter region (∼84%GC) and APOE (75.5% GC) gene of Homo sapiens mediated by Ex Taq DNA polymerase within the appropriate concentration range. We found that a combination of DMSO and glycerol additives was critical to obtaining the target amplicons. Thus, the solvents mixed with 3%DMSO and 5%glycerol were used in bismuth-based materials. This allows for better dispersion of bismuth subcarbonate. As for the enhanced mechanisms, the surface interaction of PCR components including Taq polymerase, primer and products with bismuth-based materials may be the main reason. The addition of materials can reduce the melting temperature (Tm), adsorb polymerase and modulate the amount of active polymerase in PCR, facilize the dissociation of DNA products, and enhance the specificity and efficiency of PCR. This work provided a class of candidate enhancers for PCR, deepened our understanding of the enhancement mechanisms of PCR, and also explored a new application field for bismuth-based materials. However, in my opinion, the MS needs major revisions. I have some suggestions to improve this manuscript: 

1.    The topic is relevant and interesting. The biggest problem I had with the paper was the poor English language, which made the text difficult to understand. The paper needs to go through extensive language editing before it can be considered.

2.    The introduction is not properly contextualized. It needs great improvement in explaining both merits and demerits of invasive species and methods of identification.

3.    Figures 12 and 13 have quite low resolution and are difficult to make out. Further, figure texts are not readable. Higher-resolution versions will be needed for publication,

4.    In Material and Methods:- indicate how many replicates assayed in each analysis/parameter. The number of samples or biological and technical replicates should be mentioned for each parameter in the methods.

5.    Results must be explained clearly and in detail.

6.    The discussion should be interpreted with the results as well as discussed in relation to the present literature.  

7.    References: shall have to correct the whole References according to the ”Instructions for the Authors”, e.g. the Journal name must be abbreviated, the journal name in italics, the year must be bold and the author shall have to use the without italics paper titles.

8.    The conclusion is very lengthy and requires improvement.

  Best wishes and thank you for the opportunity.

To,

The Editor,

Molecules, MDPI,

Manuscript ID: molecules-2345293

Subject: Submission of comments of the manuscript in “Molecules"

Dear Editor Molecules, MDPI,

Thank you very much for the invitation to consider a potential reviewer for the manuscript (ID: molecules-2345293). My comments responses are furnished below as per each reviewer’s comments. 

In the reviewed manuscript, the authors' two bismuth-based materials that are inexpensive and readily available were used to optimize GC-rich PCR. The results demonstrated that ammonium bismuth citrate and bismuth subcarbonate were effective in enhancing PCR amplification of the GNAS1 promoter region (∼84%GC) and APOE (75.5% GC) gene of Homo sapiens mediated by Ex Taq DNA polymerase within the appropriate concentration range. We found that a combination of DMSO and glycerol additives was critical to obtaining the target amplicons. Thus, the solvents mixed with 3%DMSO and 5%glycerol were used in bismuth-based materials. This allows for better dispersion of bismuth subcarbonate. As for the enhanced mechanisms, the surface interaction of PCR components including Taq polymerase, primer and products with bismuth-based materials may be the main reason. The addition of materials can reduce the melting temperature (Tm), adsorb polymerase and modulate the amount of active polymerase in PCR, facilize the dissociation of DNA products, and enhance the specificity and efficiency of PCR. This work provided a class of candidate enhancers for PCR, deepened our understanding of the enhancement mechanisms of PCR, and also explored a new application field for bismuth-based materials. However, in my opinion, the MS needs major revisions. I have some suggestions to improve this manuscript: 

1.    The topic is relevant and interesting. The biggest problem I had with the paper was the poor English language, which made the text difficult to understand. The paper needs to go through extensive language editing before it can be considered.

2.    The introduction is not properly contextualized. It needs great improvement in explaining both merits and demerits of invasive species and methods of identification.

3.    Figures 12 and 13 have quite low resolution and are difficult to make out. Further, figure texts are not readable. Higher-resolution versions will be needed for publication,

4.    In Material and Methods:- indicate how many replicates assayed in each analysis/parameter. The number of samples or biological and technical replicates should be mentioned for each parameter in the methods.

5.    Results must be explained clearly and in detail.

6.    The discussion should be interpreted with the results as well as discussed in relation to the present literature.  

7.    References: shall have to correct the whole References according to the ”Instructions for the Authors”, e.g. the Journal name must be abbreviated, the journal name in italics, the year must be bold and the author shall have to use the without italics paper titles.

8.    The conclusion is very lengthy and requires improvement.

  Best wishes and thank you for the opportunity.

Author Response

To Reviewer #3:

  1. The topic is relevant and interesting. The biggest problem I had with the paper was the poor English language, which made the text difficult to understand. The paper needs to go through extensive language editing before it can be considered.

Reply: Thanks for your suggestion. We have gone through extensive language editing for the paper.

  1. The introduction is not properly contextualized. It needs great improvement in explaining both merits and demerits of invasive species and methods of identification.

Reply: In order to make the introduction part more contextual, we have enriched the merits and demerits of bismuth compounds, as well as research methods. (see page 2, section 1)

  1. Figures 12 and 13 have quite low resolution and are difficult to make out. Further, figure texts are not readable. Higher-resolution versions will be needed for publication.

Reply: We have provided higher resolution and clearer illustrations of Fig. 10 and 11 (original Fig. 11 and 12) (see pages 16-17, section 3.4.3)

  1. In Material and Methods: - indicate how many replicates assayed in each analysis/parameter. The number of samples or biological and technical replicates should be mentioned for each parameter in the methods.

Reply: Thanks a lot for your reminder. We have mentioned the number of replicates two or three times in the experimental method. (see pages 3, section 2.1)

  1. Results must be explained clearly and in detail.

Reply: Thank you for your helpful suggestion. We have carried out a more accurate and argumentative description analysis of all the results, and we hope you will feel that the logic is clear when you read it (see pages 5-17, section 3)

  1. The discussion should be interpreted with the results as well as discussed in relation to the present literature.

Reply: According to your good advice, we carefully analyzed the results and discussed them about the present literature. (see pages 5-17, section 3, References 36, 44-48 in red)

  1. References: shall have to correct the whole References according to the “Instructions for the Authors”, e.g. the Journal name must be abbreviated, the journal name in italics, the year must be bold and the author shall have to use the without italics paper titles.

Reply: Thank you for your careful correction. We have corrected the References according to the “Instructions for the Authors”. (see pages 18-20, References 1-50)

  1. The conclusion is very lengthy and requires improvement.

Reply: According to your kind advice, the conclusion has been reduced to three short paragraphs, mainly divided into research summary and outlook. (see page 17, section 4)

Round 2

Reviewer 2 Report

I think the manuscript improved a lot, therfore I suggest you to accept it. 

Reviewer 3 Report

Dear Chief Editor,

Thank you for providing the opportunity to review the revised manuscript. The manuscript is improved considerably after revision according to the reviewer's comment. Now this study is a suitable contribution to the Molecules. I recommend the manuscript for publication.

Thank you

With best regards

Dear Chief Editor,

Thank you for providing the opportunity to review the revised manuscript. The manuscript is improved considerably after revision according to the reviewer's comment. Now this study is a suitable contribution to the Molecules. I recommend the manuscript for publication.

Thank you

With best regards